# Association between in situ ventilation and human-generated aerosol exposure in meatpacking plants during the COVID-19 pandemic

Joshua L. Santarpia[1,2,3☯]*, Josephine Lau[4☯], Debayan Shom[4], Shanna A. Ratnesar-Shumate[1¤], Eric C. Carnes[2,5], George W. Santarpia[5], Vicki L. Herrera[1], Danielle N. Rivera[2], Daniel N. Ackerman[1,3], Ashley R. Ravnholdt[1,2], John J. Lowe[2,5], Athena K. Ramos[6☯]

1 Department of Pathology, Microbiology and Immunology, College of Medicine, University of Nebraska Medical Center, Omaha, Nebraska, United States of America, 2 Global Center for Health Security, University of Nebraska Medical Center, Omaha, Nebraska, United States of America, 3 National Strategic Research Institute, University of Nebraska, Omaha, Nebraska, United States of America, 4 Durham School of Architectural Engineering & Construction, College of Engineering, University of Nebraska Lincoln, Lincoln, Nebraska, United States of America, 5 Department of Environmental, Agricultural and Occupational Health, College of Public Health, University of Nebraska Medical Center, Omaha, Nebraska, United States of America, 6 Department of Health Promotion, College of Public Health, University of Nebraska Medical Center, Omaha, Nebraska, United States of America

☯ These authors contributed equally to this work.
¤ Current address: Center for Aerosol Science and Technology, College of Engineering, University of Miami, Miami, Florida, United States of America
* josh.santarpia@unmc.edu

**Data Availability Statement:** Summary data may be found within the paper and/or Supporting Information files except for the COVID-19 case

## Abstract

During the COVID-19 pandemic, meatpacking workers were disproportionately affected by disease. Large outbreaks at meatpacking facilities resulted in loss of life and threatened the well-being of workers across the globe. Much work was done throughout the pandemic to understand and prevent these outbreaks. This study combined ventilation system evaluation and measurement of human-generated respiratory aerosol to investigate and identify areas of highest risk for disease transmission. These findings confirm that improved ventilation reduces exposure to human-generated aerosols in meatpacking facilities, including those that may contain infectious agents, such as SARS-CoV-2. This study suggests areas of greatest risk are likely areas where workers break from work, such as cafeterias and locker rooms, where ventilation is poorer, use of face masks is reduced, and people congregate. Furthermore, these findings also suggest that ventilation of production areas of the plant, which have been designed for food safety, is sufficient to reduce exposures and likely contributes to reduced transmission in those spaces. Based on these findings, two controls should be prioritized to minimize the likelihood of exposure to potentially infectious aerosols: (1) improving mechanical ventilation and/or adding mitigation strategies such as media filters, germicidal ultraviolet, and other air cleaning technology and (2) applying administrative practices that minimize large congregations of people in poorly ventilated spaces. Importantly, this work demonstrates a method for in situ measurements of human-generated

data, which was obtained from Center for Systems Science and Engineering (CSSE) at Johns Hopkins University (https://coronavirus.jhu.edu/region/united-states). Raw data has been deposited at: https://doi.org/10.7910/DVN/J6BQES."

**Funding:** Funding for this study was provided through private donations to the University of Nebraska Foundation. Many of the AirAnswers samplers used in this study were a gift from Inspirotec, LLC. The funders had no role in study design, data collection and analysis, decision to publish, or preparation of the manuscript.

**Competing interests:** Santarpia has been a paid consultant for both Inspirotec and Poppy Health, both of whom develop and provide indoor infectious disease and allergen air monitoring services and devices. All other authors report no conflict. The funders had no role in the design of the study; in the collection, analyses, or interpretation of data; in the writing of the manuscript; or in the decision to publish the results. Many of the AirAnswers samplers used in this study were a gift from Inspirotec, LLC. This does not alter our adherence to PLOS ONE policies on sharing data and materials.

particles that can be used more broadly to understand exposure and risk in various occupied spaces.

## Introduction and background

The meat processing industry in the United States (U.S.) employed 519,450 people in 2021 [1], and meat processing accounted for more than 20% of total county employment in 56 counties across the country [2]. The meat processing industry is dominated by global food companies such as Cargill, Tyson, JBS, and Smithfield [3], and the COVID-19 pandemic created significant challenges for the industry to maintain business operations. Meat processing facilities across the world, both large and small, in countries such as the United States (U.S.), Brazil, Canada, Australia, Ireland, Spain, Germany, the United Kingdom and France experienced significant COVID-19 outbreaks [4–9]. During the first year of the COVID-19 pandemic, at least 59,000 workers at major meat processing facilities in the U.S. contracted COVID-19 and 269 workers died from the virus [10]. These numbers likely underestimate the full impact of COVID-19 on this workforce in the U.S. since national surveillance data is not available.

This industry is especially susceptible to COVID-19 and any airborne pathogens as the production process requires a significant number of workers per shift to operate, especially in large facilities. Often, hundreds of people may be on the same shift, making these workplaces quite dense. Workers may be in close physical proximity on production lines and share common welfare spaces, including cafeterias, locker rooms, and bathrooms, which make distancing challenging [11, 12]. Many meat processing facilities throughout the U.S. are older and may not have the mechanical systems needed to ensure safe working conditions in the context of a pandemic. Additionally, the meat processing workforce in the U.S. is culturally and linguistically diverse, making communication difficult and creating challenges for implementing common public health and infection prevention strategies. As such, meat processing facilities became vectors of community transmission [13], and it has been estimated that 334,000 COVID-19 infections could be associated with meat processing facilities in the U.S. during the first year of the pandemic [8].

Although research has now proven that COVID-19 spreads both through droplet and aerosol transmission [14–17], there was much debate early on during the pandemic regarding effective measures to prevent transmission of the virus. Infection prevention and control guidance was focused on the hierarchy of controls to reduce transmission within the meat processing industry [18]. Because so little was known, most interventions centered on administrative controls and personal protective equipment [19]. Engineering controls such as changes to ventilation systems including increasing clean air flow and enhancing filtration, could be used to reduce exposure to the SARS-CoV-2 virus. However, little research has been conducted to understand airborne infectious disease transmission risks in meat processing facilities and how best to mitigate these risks.

### Ventilation and mechanical systems in meat processing facilities

Ventilation systems for meat processing facilities are typically focused on food safety as the primary goal. Ventilation guidelines for the health and comfort of workers and other occupants, including target ventilation rates, are not available [20, 21].

Prior to the COVID-19 pandemic, guidelines and recommendations suggested that filtered outdoor air should be moved from the packaging area to the processing area and then to the

areas where raw materials are handled as illustrated in Fig 1. Generally, filtered outdoor air should be moved into the cleanest spaces first and then flow out to other spaces. This design is intended to maintain the highest positive pressure so that the product is less likely to be exposed to contaminants from the outdoor air. Bringing unfiltered outdoor air into the indoor space is discouraged as it is a potential source of contamination from bacteria, pollen, and other particles. Positive pressurization is generally suggested for the entire facility, and appropriate filtration and frequent cleaning of outdoor air intake units used for pressurization should be conducted to avoid airflow interruptions [20].

Although the United States Department of Agriculture (USDA) Facility Guidelines for Meat Processing Plants states, "There should be enough ventilation for all areas of the establishment including workrooms, processing, packaging, and welfare rooms to ensure sanitary conditions" ([21], p. 45032), no specific design guidance is provided for the exact amount of fresh air that should be provided during normal working conditions. In the context of the pandemic, even less about ventilation rates is known. Finci et al. [22] presented different zone arrangements for ventilation and mechanical systems and the probability of SARS-CoV-2 infection in a meat processing facility early in the pandemic; however, actual ventilation rates in different zones and in-situ settings of those mechanical systems were not measured. In another study, which collected data from 22 meat processing plants in Germany, there were fewer COVID-19 cases in well-ventilated spaces, suggesting that increasing ventilation could mitigate the risk of infection, but in-situ ventilation rates were not measured for this study either [23]. Other studies mention the need for improved ventilation but lack detail on specific ventilation rates that could be considered sufficient [11, 24]. As evidenced, there is a clear gap in knowledge on the in-situ ventilation rates in meat processing facilities and their association with COVID-19 infection risk. Data is needed to inform recommendations on appropriate ventilation rates to reduce airborne pathogens and related infection risks in meat processing facilities.

## Objectives

This study aimed to evaluate the heating, ventilation, and air conditioning (HVAC) systems to measure airflow patterns and the in-situ ventilation rates in meat processing plants and to assess exposure to SARS-CoV-2 and other human-generated aerosol particles in air samples throughout three meat processing facilities. This data allows exploration of the relationship between measured ventilation rates and exposure to human-generated aerosols, including those containing SARS-CoV-2.

## Methodology

### Site and experimental description

Data was collected from three different meat processing facilities with unique HVAC configurations, namely Site A, Site B, and Site C, in this study (Tables 1 and 2). Two of these sites (A and C) were beef plants and one (Site B) was a poultry processing facility. The plant structure was generally divided into kill/slaughter areas (harvest area), processing and packing areas, cafeterias and common welfare areas (e.g., locker rooms, bathrooms, and general office/meeting rooms). Air sampling studies (Table 2) were performed at Site A between April 13–15, 2021, at Site B between September 13th and September 22nd, 2021, and at Site C between August 3–5, 2021. These time periods corresponded to the dominance of different SARS-CoV-2 variants of concern and different levels of community spread (Fig 2).

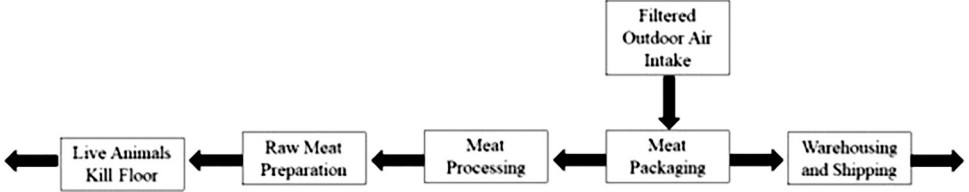

**Fig 1. Diagram of air pressure and flow directions in typical meat processing facilities (adapted from [20, 21]).**

## Preliminary study at Site A

For Site A only, preliminary studies began the week of March 3, 2021 immediately following a small COVID-19 outbreak of approximately 25 cases among workers that was identified the week of February 22, 2021. Preliminary surface samples were collected during the first site visit on March 3, 2021. Long-term air sampling devices (see below) were initially deployed at this time and were subsequently recovered on March 11, 2021 and March 17, 2021 for analysis. Beginning on March 3, 2021, Site A began a vaccination campaign that resulted in 74.41% of staff vaccinated with one dose and 61.91% of staff being fully vaccinated by April 13, 2021.

## Evaluation of the HVAC system—Test 1, 2, and 3

The research team conducted field measurements and inspections to evaluate the performance of the ventilation systems in each facility. Air circulation and ventilation rates in the selected areas of each site were measured using the methods described in the following three tests:

1. Carbon dioxide ($CO_2$) based tracer gas method

2. Airflow rate measurements from the diffusers

3. $CO_2$ measurements at the intake of roof top ventilation units

**Test 1 –Carbon dioxide ($CO_2$) based tracer gas method.** Test 1 is the $CO_2$-based tracer gas measurement method derived from American Society for Testing and Materials (ASTM) standard D6245 [25], which is a well-developed method to estimate the in-situ ventilation rate. Because this method applies to single-zone systems, spaces were sealed and isolated. Given the need to seal the spaces, this test was done during periods when workers were not present, but the ventilation system was operating as normal. Then, the research team used multiple $CO_2$ fire extinguishers to raise the $CO_2$ concentration to a target of 5,000 to 8,000 ppm. Decay methods were employed to calculate the ventilation rates from the decay curves.

**Table 1. Information related to the mechanical systems and its operating conditions for different areas at each plant, including filter ratings used in the systems and reported as Minimum Efficiency Reporting Values (MERV).**

|  | Site A | Site B | Site C |
|---|---|---|---|
| Processing areas | The fabrication area is pressurized by outdoor air | Only recirculation | Only recirculation |
| Kill (Slaughter) areas | Exhaust air only | Exhaust + make-up outdoor air | Exhaust + make-up outdoor air |
| Common areas | Only recirculation | 15% outdoor air (with economizer) | Only recirculation |
| Additional filters/cleaner (in both processing and common areas) | MERV 13 or 16 In-duct GUV (Germicidal Ultraviolet), portable ionizer in some common areas | Upgraded to MERV 13 right before the site measurement | MERV 4 or 8 filters in-unit UV oxidizing air purifying devices |

**Table 2. Sampling and background information for each plant location.**

|  | Site A | Site B | Site C |
| --- | --- | --- | --- |
| Harvest area samples | 4 per day | 4 per day | 10 per day |
| Processing and packing area | 12–15 per day | 12 per day | 10 per day |
| Cafeteria | 4–5 per day | 4 per day | 5 per day |
| Common areas | 5 per day | 4 per day | 5 per day |
| Long-term samples | 6 total | 6 total | 6 total |
| Number of employees | ~3600 | ~1000 | ~3500 |
| Largest COVID-19 outbreak cluster | 237 | 110 | 264 |
| Percent vaccinated employees at the time of sampling | 62% | 62% | 59% |

To calculate the air change per hour (ACH), the research team plotted $CO_2$ reading, ppm–outdoor air $CO_2$, ppm and then used exponential decay curve fitting to find the coefficients for the exponential curve. The equations are in the form of $y = Ae^{-Bx}$. The negative sign is for the exponential decay. The value of B provides the air change per 5 minutes. So, in order to get the ACH, B is divided by (5/60). The dimensions of the space were measured, and the volume (V) was calculated for all locations. Q, ventilation rate in ft$^3$/min was calculated using the formula:
$Q = \frac{ACH*V}{60}$

**Test 2 –Airflow measurement from diffusers.** Test 2 was used to measure the total airflow and distribution in various rooms. This test was carried out using an airflow capture hood, Model 420 by Testo (https://www.testo.com/en-US/testo-420/p/0563-4200), when the ventilation system was operating as normal during regular production operations. The actual supply flow rates through each diffuser were recorded by averaging five continuous readings taken at each diffuser. During these tests, the layout of the space was sketched, and airflows from all diffusers in the area were measured. Airflow distribution was evaluated based on the in-situ measured airflow and the diffuser location in those areas. The measured diffuser flow

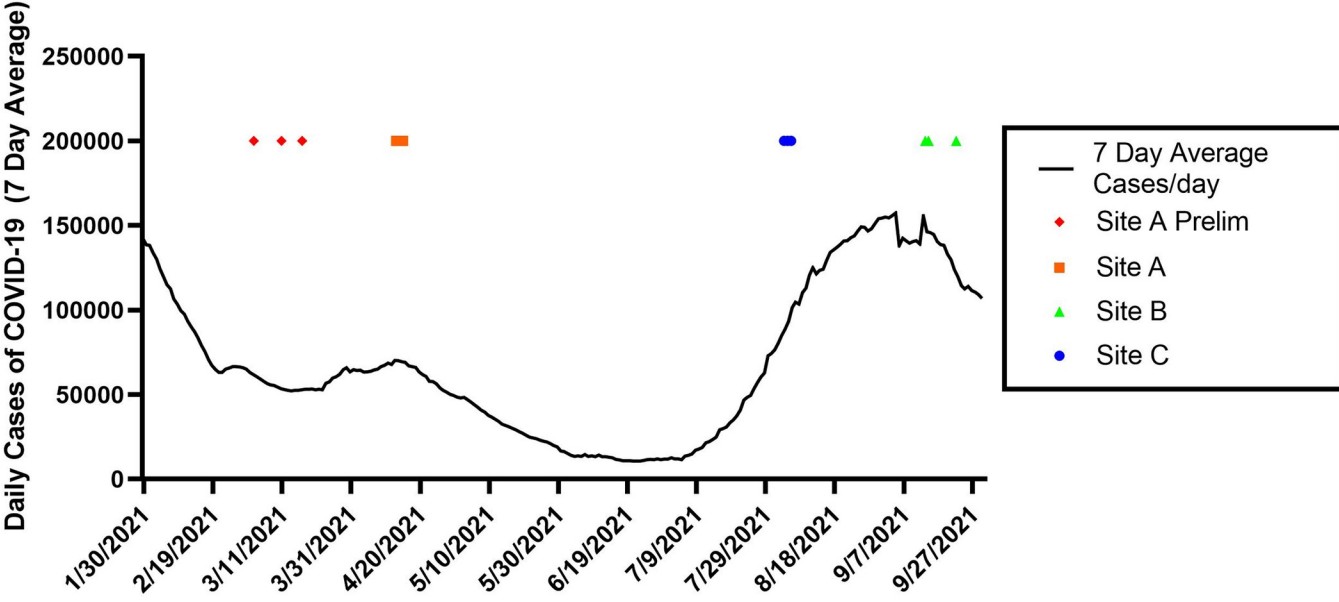

**Fig 2.** National 7-day average daily case rates (solid line) obtained from the Center for Systems Science and Engineering (CSSE) at Johns Hopkins University (https://coronavirus.jhu.edu/region/united-states) during sampling for sites A, B and C (noted with colored indicators).

rates in office spaces and the common welfare areas were selected based on each facility's layout, and the research team obtained the design airflow rates from each site's facility management team.

**Test 3 –CO$_2$ measurement in the roof-top units (RTUs).** For Test 3, the research team installed CO$_2$ monitors at the recirculation path of the RTUs in order to monitor the CO$_2$ profiles over both occupied (regular production operations) and unoccupied time to estimate ventilation rates in some of the common areas. The CO$_2$ concentration of the return air was measured and was assumed to be the same as the space concentration. The team also assumed that occupants were the only source of CO$_2$ in these measured common areas. Test 3 was not carried out in any production areas because of the usage of dry ice in those areas. The calculation and curve fitting procedure were similar to that in Test 1. Ventilation rates were obtained from decay curves of five consecutive days to ensure the repeatability of the measurements and a more reliable estimation of the actual ventilation rates.

## Air and surface sample collection

A component of this study was to perform air sampling to identify markers of respiratory transmission of diseases (including evidence of SARS-CoV-2) in production areas where workers were in close proximity and in common areas throughout the plant where large numbers of workers may congregate such as locker rooms, equipment distribution areas, hallways, and cafeterias. This work was broken down into three discrete sampling modalities:

1. Long-term sampling

2. Discrete-term sampling

3. Surface sampling

**Test 4 –Long-term air sampling.** Continuous sampling was performed in six or seven locations identified as high-traffic areas for employees and in which no harsh decontamination processes were implemented that might degrade the air sampling equipment or air samples overnight at each site (Table 1). AirAnswers (Inspirotec1, Chicago, Il., USA) passive air sampling devices were installed upon arrival of the research team on the first day at each site and retrieved at the end of sampling activities (typically 3 days later). For recovery, cartridges were removed from the sampler and placed in a sealed bag for transport. The two metallic probes were removed from each cartridge and placed in a 15 mL conical tube with 10 mL phosphate buffered saline (PBS) and then shaken by hand for one minute to liberate collected particles.

**Test 5 –Discrete-term air sampling.** Discrete air sampling was performed at each site over multiple locations and times that were identified as those in which there was high-traffic and close proximity among employees at each site (Table 2). Some samplers were placed amid workers and should be representative of personnel exposure, while others were placed at the perimeter of work areas or on the catwalks and might be more representative of the general area. These samples were taken at times of maximum occupancy in each area. Sartorius MD8 Airscan® (https://www.sartorius.com/s) were utilized to collect air samples for 30-minute intervals. Each sampler ran on battery power and was operated manually throughout the duration of sampling. After each 30-minute sampling period, each of the Sartorius MD8 Airscan® samplers was turned off, and the samples recovered for subsequent assay. After recovery, the samplers were moved to the next location for sampling as listed in Table 2 and the process repeated. Sartorius filters were recovered by dissolving each gelatin sampling filter in 10 mL of 1x PBS (pH 7.2; Gibco, 70010–023) pre-warmed at 37˚C.

**Test 6 –Surface sampling.** Each surface sample was collected using a 3" x 3" sterile gauze pad placed in a 50 mL conical tube and presoaked with 3 mL of 1x PBS. To capture potential SARS-CoV-2 surface contamination, the gauze was removed from the conical tube, swiped on the HVAC grate in a double "s" pattern, returned to the conical tube, and placed on ice to preserve viral infectivity and the integrity of viral genomic material before processing, which was generally performed the same day as sample collection, but no longer than 24 h post collection. Samples were recovered using 5 mL of 1x PBS added to each 50 mL conical tube. The conical tubes were shaken by hand for approximately 30 seconds, and an aliquot of the extracted viral suspension was used for ribonucleic acid (RNA) extraction, as described below.

**Assay of samples.** Qiagen EZ1 Advanced XL instruments paired with Qiagen Virus Mini Kits v2.0 (QIAGEN GMbH, Hilden, Germany) were used to extract RNA from the samples. Samples were eluted in 60 µL of Qiagen Buffer AVE. Following extraction, samples were analyzed for SARS-CoV-2 RNA using a quantitative reverse transcription-polymerase chain reaction (qRT-PCR) assay targeting the E gene of SARS-CoV-2 [14]. As a control, unused presoaked gauze and dissolved gelatin filters were analyzed in addition to the sample RNA. The qRT-PCR assay used the Invitrogen Superscript III Platinum One-Step RT-PCR System. Each PCR run included a viral RNA positive control and a nuclease-free water aliquot as a negative control. Samples were run in triplicate. A cycle threshold (Ct) of 39 or lower on any of the three replicates would be considered a positive detection. Reactions were run with initial conditions of 10 min at 55˚C and four min at 94˚C, and then 45 cycles of 15 seconds at 94˚C and 30 seconds at 58˚C. The target sequences for the E gene were as follows:

Probe: 5′/56-FAM/ACACTAAGCC/ZEN/ATCCTTACTGCGCTTCG/3AIBkFG/−3'
Primer 1: 5′-ATATTGCAGCAGTACGCACACA-3'
Primer 2: 5′−ACAGGTACGTTAATAGTTAATAGCGT-3'

In addition, RNA extracted from the samples was analyzed for the presence of human surfactant protein C (SFTPC). SFTPC is expressed in Type II alveolar cells in the lung and is not known to be produced in other tissues [26], and the presence of this mRNA as indicator of human respiratory material. The assay was designed to use the same amplification conditions as the SARS-CoV-2 E gene assay so that they could be run simultaneously. The primers and probe for SFTPC (Genbank accession number NM_003018.4) were as follows:

Probe: 5'-/56-FAM/AGCATAGTG/ZEN/AGGTGGACAGCTAGTACC/3IABkFQ/−3'
Primer 1: 5'-CGTTCTGCTTTGTTAGGCATTAG-3'
Primer 2: 5'-TGTCACACCCATGATGCTATT-3'

A Primer-BLAST search [27] of this primer combination indicates that in addition to human, some cross-reactivity may be observed in Bonobo and African Green monkey sequences. Both E gene and SFTPC assays were run in triplicate for each sample. Amplification at 39 Ct or lower, for SARS-CoV-2, or 45 or lower for SFTPC on any one of the triplicate runs was considered a positive detection.

A standard curve run in triplicate using synthetic DNA was used to quantify viral RNA and SFTPC mRNA from each sample using Ct obtained from RT-qPCR. The data were fit with the following exponential functions:

$$\frac{\textbf{SARS}-\textbf{CoV}-\textbf{2 copies}}{\textbf{mL}} = \textbf{9.0} \times \textbf{10}^{12}\textbf{e}^{-0.554*\textbf{Ct}}$$

$$\frac{\textbf{SFTPC copies}}{\textbf{mL}} = \textbf{1.0} \times \textbf{10}^{15}\textbf{e}^{-0.701*\textbf{Ct}}$$

The value of each triplicate Ct value was used in the equation above to calculate average copies/mL. Undetected samples were evaluated at zero (copies/mL) before calculating the

**Table 3. Site A production space ventilation rates measured by Test 1.**

| Space Name | Max. Occupancy | Space Volume (cubic ft.) | Air changes per hour (ACH) | Measured ventilation (cfm/person) |
|---|---|---|---|---|
| Fabrication area | 170 | 18,9196 | 8.84 | 114.5 |
| Production area: Offal | 30 | 38,402 | 6.31 | 122.6 |

average concentration. Total copies collected, copies/cm$^2$ and copies per liter of air were then calculated from the liquid recovery volumes, surface areas sampled, and volume of air sampled for each sample.

**Environmental characterization.**   In addition to the air sampling instruments described above, the AZ 7755 (AZ Instrument Corp., Taiwan, R.O.C.) was used to measure $CO_2$ concentrations in all areas where air samples were collected. Recording of these environmental variables was concurrent with all collected air samples.

**Data tabulation and analysis.**   Raw data from all experiments were tabulated and reported parameters were calculated, according to the equations described, along with the associated means and standard deviations across multiple samples using Microsoft Excel v2410. Further statistical analysis, including correlations, regressions, and significance calculations were performed using GraphPad Prism v9.5.1.

## Results

### Evaluation of the HVAC system

The ventilation rates of different spaces are compared with similar occupancy types listed in American Society of Heating, Refrigerating and Air-Conditioning Engineers (ASHRAE) standard 62.1–2022 [28]. For the production spaces (i.e., kill/harvest and fabrication), we were only able to use Test 1 (the tracer gas method) to measure the ventilation rate in Site A during an extended unoccupied (i.e., non-production) time. The measured ventilation rates in these production areas are presented in Table 3.

Test 1 was also conducted in common welfare areas of Sites A and B (Table 4). The team found under-ventilated areas for Site A, while Site B had well-ventilated areas. The ASHRAE-required ventilation rates were calculated based on the breathing zone requirement stated in ASHRAE standard 62.1–2022 [28] by adding the "people outdoor air rate" and "area outdoor air rate" components. The research team also performed Test 3 in some of the common areas

**Table 4. Ventilation rates in common spaces measured by Test 1 and Test 3.**

| Site | Test type | Space Name | Area (sq. ft.) | Max. Occupancy | Measured ventilation (cfm/person) | ASHRAE Ventilation requirement (cfm/person) |
|---|---|---|---|---|---|---|
| Site A | Test 1 | Cafeteria I | 2,784 | 280 | 2.80 | 9.94 |
| | Test 1 | Locker Room II | 3,210 | 617 | 0.61 | 5.31 |
| | Test 3 | Cafeteria I | 2,784 | 280 | 2.49 | 9.94 |
| | Test 3 | Office space | 3,060 | 15 | 13.67 | 17.24 |
| Site B | Test 1 | Training classroom | 1,963 | 66 | 18.36 | 6.79 |
| | Test 1 | Cafeteria II | 4,466 | 290 | 5.20 | 10.27 |
| | Test 3 | Locker Room | 1,642 | 22 | 15.76 | 9.48 |
| | Test 3 | Cafeteria I | 7,865 | ~100* | 16.94 | 21.66 |
| Site C | Test 3 | Cafeteria II | 4,675 | 212 | 3.4 | 11.47 |
| | Test 3 | Office space | 1,440 | 12 | 11.66 | 12.2 |

*Approximated based on actual worker shift schedule observed by research team

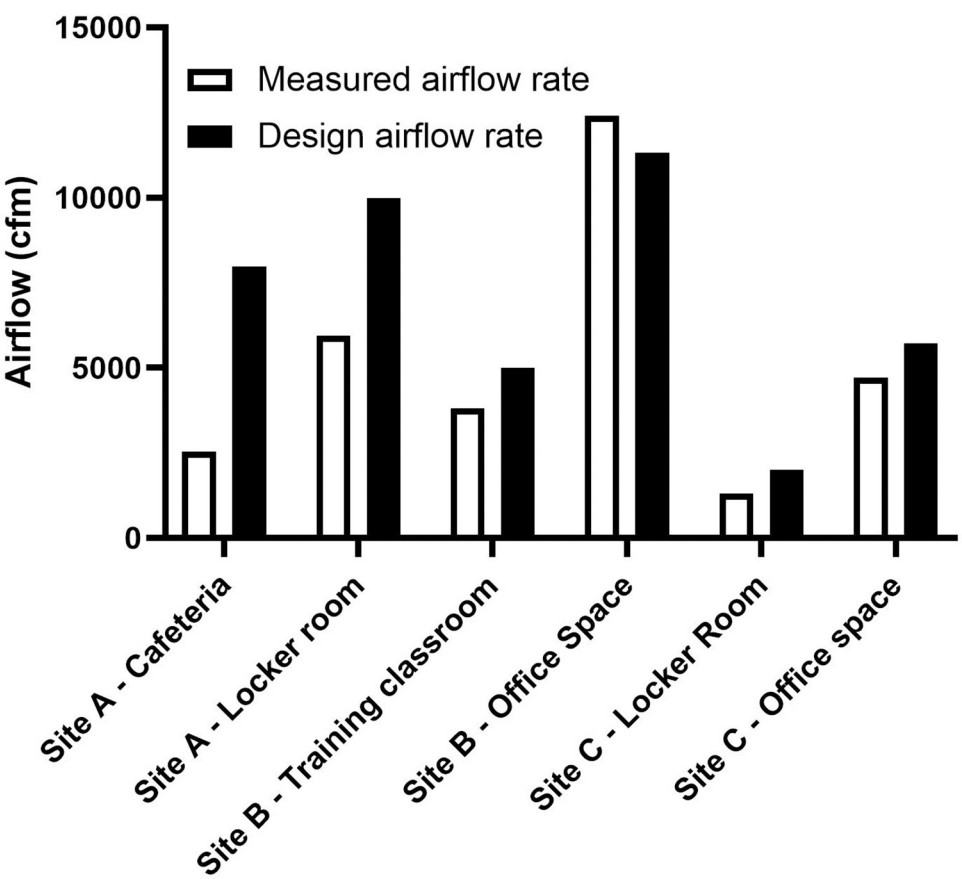

**Fig 3. Airflow from diffusers for selected spaces.**

already measured in Sites A and B while including additional common spaces in site C. Overall, Site B was found to be well-ventilated in almost all the locations where measurements were taken, and Site A and Site C had a multiple under ventilated locations.

Based on Test 2, some of the office spaces and common welfare areas did not have a uniformly distributed airflow from the diffusers and the total flow rate from these diffusers were less than the design flow rates (Fig 3).

## Surface and air sampling

**Site A.** During the initial site A visit on March 3, 2021, surface samples were collected in twelve locations and were assayed to determine if any RNA from SARS-CoV-2 could be detected. Surface samples were collected from seven locations in the harvest area, two locations in cafeterias, and three locations in common areas. Surface samples obtained from the boot wash wall in the Fabrication Area and from the Men's Locker Room were positive for RNA from SARS-CoV-2. All other samples were negative for RNA from SARS-CoV-2 (Table 4).

Long-term air samplers were placed in seven locations on March 3, 2021, including three in cafeterias and four in common areas. Samples were recovered on March 11, 2021 and March 17, 2021. As shown in Table 5, two air samples were positive for RNA from SARS-CoV-2 from the samples recovered on March 11, 2021: one in the main hallway personal protective equipment (PPE) check-out point and another in a conference room. No SARS-CoV-2 RNA was detected in any of the samples recovered on March 17, 2021.

**Table 5. Results from preliminary surface and long-term sampling at Site A.** NT indicate sample not taken.

| Site A | Number of Samples | SARS-CoV-2 Detected | | SARS-CoV-2 Concentration | | Number of Samples | SARS-CoV-2 Detectected |
|---|---|---|---|---|---|---|---|
| | | 3/3/21-3/11/2021 | | | | 3/11/21-3/17/2021 | |
| Harvest Areas (copies/cm$^3$) | 7 | 1 sample | mean | $1.4 \times 10^2$ | | NT | NT |
| | | 14% | std. dev. | $2.5 \times 10^2$ | | NT | NT |
| Cafeterias (copies/cm$^3$) | 2 | 0 samples | mean | NA | | NT | NT |
| | | 0% | std. dev. | NA | | NT | NT |
| Common Areas (copies/cm$^3$) | 3 | 1 samples | mean | $4.1 \times 10^3$ | | NT | NT |
| | | 33% | std. dev. | $2.8 \times 10^3$ | | NT | NT |
| Long Term Samples (toal copies) | 7 | 2 samples | mean | $5.1 \times 10^4$ | $8.9 \times 10^4$ | 7 | 0 samples |
| | | 29% | std. dev. | $2.2 \times 10^4$ | $3.8 \times 10^4$ | | 0% |

Long-term sampling was also performed over the course of the comprehensive sampling study in six locations that were identified as high-traffic areas for workers. Samplers were installed upon arrival on April 13, 2021, the first day of sampling. These samplers were placed

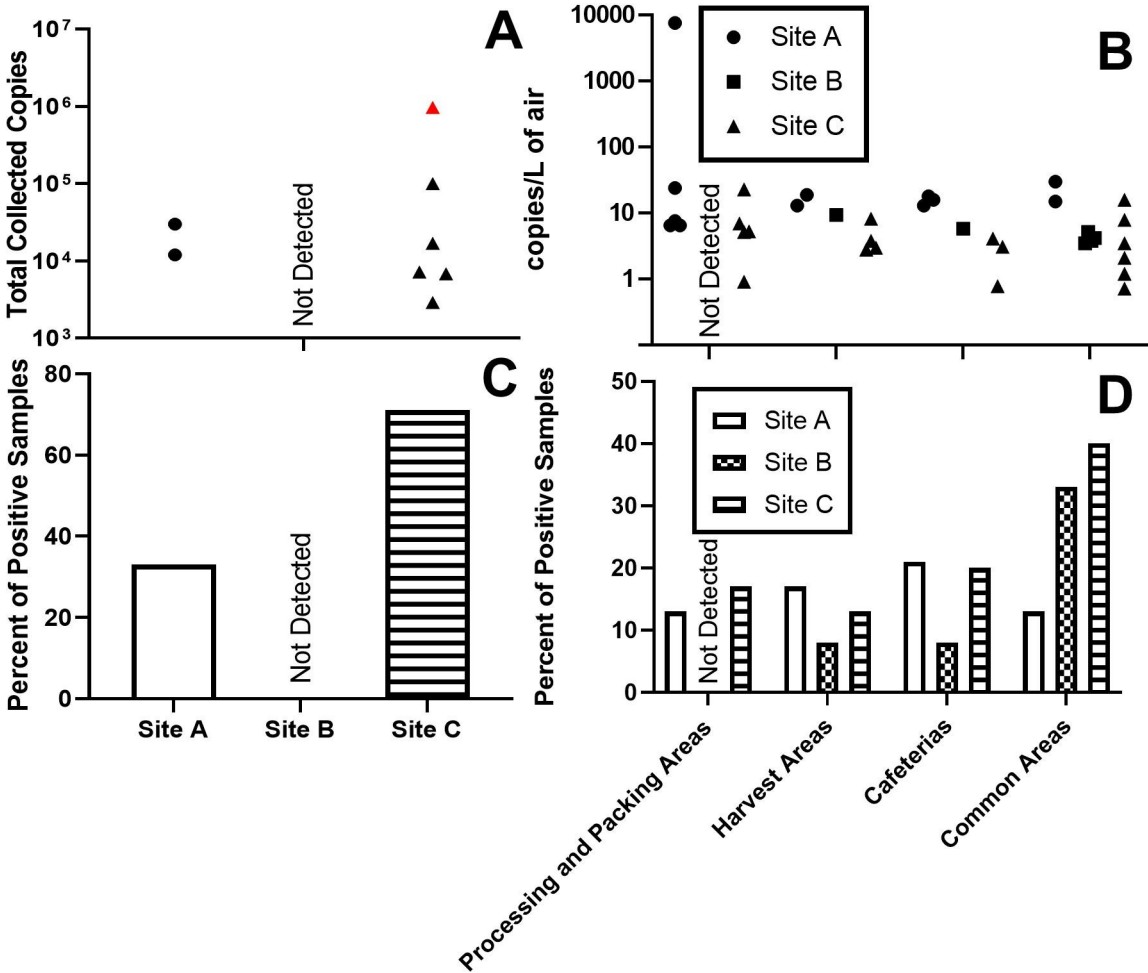

**Fig 4.** Results of long-term aerosol sampling (A and C; Test 4) and discrete-term aerosol sampling (B and D; Test 5). Data shown in S1–S3 Tables. SARS-CoV-2 was detected in these samples. The single detection of SARS-CoV-2 in the long term samples is shown in red.

in a men's locker room, women's locker room, occupational health office waiting room, main hallway PPE check-out point, and two cafeteria areas. On the final day of the study, April 15, 2021, each sampler was recovered for subsequent assay. No SARS-CoV-2 RNA was detected in any of the long-term air samples (S1 Table). The human respiratory surfactant RNA was detected in the long-term air samples from the fabrication area men's locker room and one of the cafeteria samples (Fig 4A and 4C, S1 Table).

Samples were collected in each location on April 13th (Day 1), 14th (Day 2), and 25th (Day 3) of 2021. Four to five samplers were distributed throughout each of the large processing and packing areas of the plant. None of these samples were positive for SARS-CoV-2 during the comprehensive sampling campaign period, but SARS-CoV-2 was detected in common spaces in the preliminary samples. Human generated aerosol (SFTPC RNA) was most frequently detected in the cafeterias (21% of cafeteria samples, Fig 4B and 4D, S1 Table).

**Site B.** Long-term air samplers were placed in seven locations on September 13th, 2021 and allowed to sample continuously until September 17, 2021. These samplers were placed in the office area, main plant hallway, main plant entry, men's locker room, women's locker room, and cafeterias. The samples were recovered on September 17, 2021. As shown in Fig 4A and 4C (S2 Table), no RNA from SARS-CoV-2 or the respiratory surfactant protein was detected in any of the long-term samples.

Samples were collected in each location on September 13th (Day 1), 15th (Day 2), and 22nd (Day 3) of 2021. Samplers were distributed throughout each of the large areas of the plant (i.e., harvest and processing areas). In the case of the harvest areas, fewer samplers were needed due to the limited number of workers and physical arrangement of the space. In the processing area, four samplers in three stages for a total of 12 samples were used to characterize the space each day. Ten of the 16 samples taken in these areas each day were taken amid workers, while the other six were taken around the perimeter or in minimally occupied spaces (e.g., packing).

An additional eight samples per day were taken in the common spaces (e.g., cafeterias, locker rooms, common hallways). No RNA from SARS-CoV-2 was detected in any of the samples (Fig 4B and 4D, S2 Table); however, respiratory surfactant protein was measured at various locations (Fig 4B and 4D, S2 Table). Human respiratory surfactant protein RNA was consistently measured during discrete-term sampling (in two out of the three days) in the main plant hallway and was also measured in the men's locker room, the women's locker room, the cafeteria, and the harvest area, accounting for 83% of the detections. Only one detection of human generated aerosol was made in a work area.

**Site C.** Long-term air samplers were placed in seven locations on August 2, 2021 and allowed to sample continuously until August 5, 2021. These samplers were placed in the occupational health office, two cafeterias, main entryway, main hallway, a men's locker room, and a women's locker room. Samples were recovered on August 5, 2021. The sample recovered from the fabrication men's locker room was positive for RNA from SARS-CoV-2 and the respiratory tract surfactant (Fig 4A and 4C, S3 Table). Respiratory tract surfactant was also measured in the samples recovered from the fabrication cafeteria, main entryway, main hallway, fabrication men's locker, and fabrication women's locker.

Samples were collected in each location on August 3rd, August 4th, and August 5th of 2021. Four to five samplers were distributed throughout each of the large processing and packing areas of the plant. The presence of human respiratory tract surfactant protein was detected across multiple days and locations (Fig 4B and 4D, S3 Table). In 40% of the samples collected from the common areas, and in 20% of samples from the cafeterias, RNA from human respiratory tract surfactant protein was present. By comparison, in the work areas, which included both harvest and processing areas, only 15% of the samples were positive for the SFTPC RNA.

## Synthesis of ventilation and aerosol measurements

It is clear from the observations that the SFTPC was more likely to be observed in common areas and that ventilation rates per person in the common areas of both Site A and C were generally poor compared to ASHRAE standards (Table 6). A visual inspection of data suggests a relationship between lower ventilation and observations of SFTPC. Two relationships stand out in comparisons between the ventilation rate and observations of SFTPC. First, there was a moderate, but significant correlation (Pearson's R of -0.55, $p = 0.03$) between the measured ventilation rate per person and the percent of SFTPC positive samples observed in each space where both measurements were made. This indicates that SFTPC is more likely to observed in areas of lower ventilation. Further, there was an exponential relationship ($R^2$ of 0.46) between

**Table 6. Lists spaces where measurements were taken and the ratios between measured ventilation rates and the typical ventilation rates based on ASHRAE Std 62.1 as well as observations of human SFTPC in those same spaces as a proxy for exposure to human generated respiratory aerosol.** The calculated mean concentration includes all measurements made in each space including observations where no SFTPC was measured. NM indicates measurements that were not made in each space.

| Site | Location | Measured ventilation (cfm/person) | Ratios of Measured ventilation rate to ASHRAE recommended ventilation rate | Ventilation Status | Percent of Discrete Aerosol Samples Positive for SFTPC | Mean Concentration of SFTPC (copies/L of air) |
|---|---|---|---|---|---|---|
| A | Fabrication area | 114.5 | NA | Good | 16.7 | 2.51 |
| | Offal area | 122.63 | NA | Good | 16.7 | 2.74 |
| | Cafeteria I | 2.8 | 0.245 | significantly under-ventilated | 21.4 | 3.34 |
| | Locker Room I | 0.61 | 0.115 | significantly under-ventilated | 33.3 | 10.02 |
| | Locker Room II | 0.43 | 0.079 | significantly under-ventilated | 33.3 | 5.15 |
| | Office space | 13.67 | 0.793 | slightly under-ventilated | NM | NM |
| B | Training classroom | 7.68 | 1.131 | Good | NM | NM |
| | Office space | 24.07 | 2.381 | Good | NM | NM |
| | Cafeteria I | 3.32 | 0.782 | slightly under-ventilated | 16.7 | 0.97 |
| | Cafeteria II | 2.42 | 0.452 | significantly under-ventilated | NM | NM |
| | Locker Room I | 16.83 | 1.874 | Good | 33.3 | 1.40 |
| | Locker Room II | 15.76 | 1.662 | Good | 33.3 | 1.28 |
| C | Cafeteria I | 1.77 | 0.157 | significantly under-ventilated | 16.7 | 0.13 |
| | Cafeteria II | 3.4 | 0.296 | significantly under-ventilated | 33.3 | 1.22 |
| | Locker I | 0.99 | 0.184 | significantly under-ventilated | NM | NM |
| | Locker II | 2.04 | 0.378 | significantly under-ventilated | 33.3 | 1.17 |
| | Locker III | 0.93 | 0.173 | significantly under-ventilated | 33.3 | 0.24 |
| | Locker IV | 2.51 | 0.457 | significantly under-ventilated | NM | NM |
| | Office space | 7.76 | 0.956 | slightly under-ventilated | NM | NM |

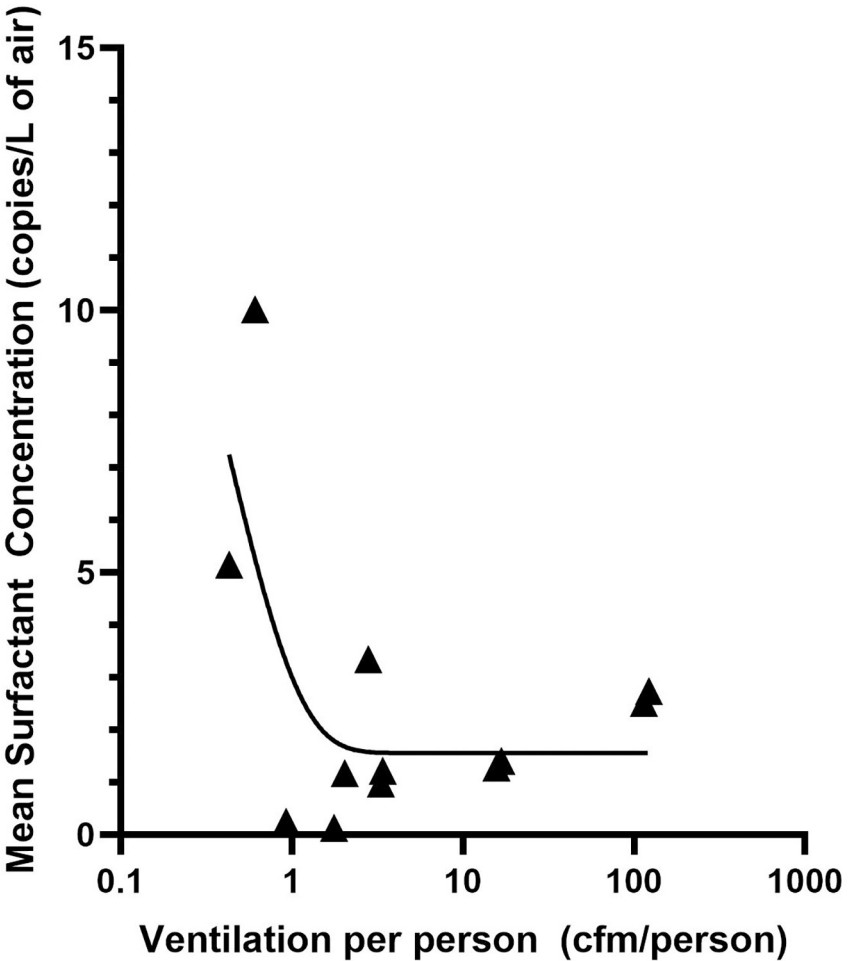

**Fig 5. The mean concentration of respiratory surfactant observed as a function of ventilation.**

the mean concentration of SFTPC and the ventilation rate (Fig 5), with the following relationship:

$$SFTPC \left( \frac{copies}{L \ of \ air} \right) = 15.89 \cdot e^{-2.387V+1.555}$$

where $V$ = ventilation (cfm/person). If measurements of SFTPC are considered a proxy for exposure to human generated aerosol, this indicates that in areas where ventilation is lower a person is more likely to be exposed to the aerosols generated by other people, and the magnitude of that exposure increases exponentially as the ventilation rate decreases. Therefore, it is not surprising that our only observations of SARS-CoV-2 aerosol (in the long-term samplers) were in locker rooms at Site A and C where the observed ventilation was the poorest (<1 cfm/person).

## Discussion and conclusion

This study represents the first comprehensive exposure/risk assessment of human-to-human disease transmission in meatpacking plants. This assessment relied on multiple approaches to evaluate both the mechanical and ventilation systems, as well as novel measurements of

exhaled human aerosol utilizing RT-PCR assays of both a human respiratory surfactant and SARS-CoV-2. The most poorly ventilated common spaces, the locker rooms at Site A and C, were the only locations where SARS-CoV-2 was found, supporting the idea that the potential risk of exposure to infectious pathogens increases in poorly ventilated spaces. The per-person ventilation rates and the fraction of samples that were positive for human respiratory proteins were inversely correlated, indicating that lower ventilation leads to a higher risk of exposure. Furthermore, there was a negative correlation between ventilation and the mean concentration of human-generated respiratory particles, which indicates that the magnitude of the exposure increases exponentially with decreasing ventilation. This represents the first in situ, experimentally derived relationship between ventilation and human-generated bioaerosol exposure determined in a manufacturing/food processing environment, and the relationships found were in general agreement with models of exposure.

The current USDA and ASHRAE standards do not include ventilation recommendations or requirements for meat processing or similar types of production. However, ASHRAE standard 62.1–2022 [28] requires average outdoor air supply rates ranging from 5 cfm/person for lobbies to 36 cfm/person for general manufacturing spaces. Based on the measurement results shown in Table 3, the ventilation rates in the production areas that were measured were higher than the highest requirements listed in that standard (110 cfm/person). These production areas have higher ventilation rates because they were designed to be positively pressurized by outdoor air. As such, the air from the surrounding spaces would not be entering these production areas (see Fig 1). Based on the results in Test 2, non-uniform airflow can lead to a non-uniform distribution of contaminants or a higher concentration of infectious particles in poorly circulated areas, and airflow less than the design value would mean lower dilution of infectious particles leading to higher probability of virus transmission [29].

It is important to note that by the time of sampling, major infection prevention measures had been implemented at the sites, such as face mask policies and vaccination campaigns, which reduced the circulation of the virus among workers in those plants. Low circulation rates of COVID-19 among workers during this period made observations of SARS-CoV-2 unlikely. Therefore, the observations of human respiratory protein and in situ measurements of ventilation rates in the mechanical systems are critical to understanding disease transmission risk from respiratory particles in these environments. The prevalence of human-generated particles in the common areas, and their comparative absence in the production areas, suggests that transmission is most likely to occur in areas where there is: (1) Limited use of face masks (such as in cafeterias and locker rooms); (2) High levels of human traffic (such as during shift changes or lunch breaks); and (3) Lower per person ventilation rates (such as in locker rooms, cafeterias, and other non-production areas).

This study suggests that transmission risk in congregate spaces can be reduced exponentially by increasing the per-person ventilation rate in under ventilated spaces. Site B had a modern HVAC system (higher per-person ventilation rate when compared to other sites) and little evidence of human-generated particles, compared to other sites. The HVAC systems in sites A and C were older (recirculating air in the common areas) and as found in the site measurements, were not functioning as designed. Additionally, despite the concerns about overcrowded conditions in the production areas, the high ventilation rates in those areas, required for food safety, reduced the potential exposure of workers, particularly when respiratory protection was worn. In some cases, increasing ventilation could be done through normal maintenance and check-ups, repair of the ventilation system, and system re-balancing. In other cases, modernization of the HVAC system would be required to improve ventilation to the degree necessary to meaningfully impact exposure risk. Although not measured directly in this study, more effectively removing particulate matter through improved filtration is another way to

potentially reduce risk. ASHRAE standard 241, released in 2023, provides guidelines for reducing infection risk in the built environment, by employing effective air cleaning or filtration. Another alternative way to improve the per person ventilation is to enact policies that reduce crowding in common spaces and limit opportunities for large congregations of people. Policies such as phased shift changes and staggered lunches and breaks could be designed to limit the number of workers present at one time in common spaces. Although Site B had existing policies for phased shift changes and staggered lunches that reduced congregation around lunches and shift changes, all sites had time periods where large groups gathered in common spaces. In particular, PPE pickup stations during shift change were by far the most crowded at every site, followed by lunch breaks in the cafeterias. Careful consideration of occupancy and ventilation rates could be used to offset some deficiencies in HVAC systems in many cases.

These are important findings for the meatpacking industry and workers, which could help minimize human-to-human disease transmission beyond the context of the COVID-19 pandemic. The findings may also be applied to help minimize the transmission of seasonal influenza and other respiratory diseases that spread in congregate settings through aerosols. The meatpacking industry and union representatives are encouraged to be open to future onsite research that may impact worker health, safety, and well-being. The relationship between ventilation and exposure is likely similar throughout built environments, not just within the meatpacking industry, given that facilities and spaces in this study had markedly different HVAC designs and equipment. The findings from this study and the relationships determined here can be applied to a variety of spaces and are not limited by contrived experiments since the data were generated during normal occupancy and operations.

The results of this work can be directly applied to reducing infection risk in meatpacking plants. Application of these data in models, such as a Wells-Riley [30], can be used to calculate the airborne infection risk to compare different engineering and administrative solutions. Infection risk modeling in common spaces can be used to rank potential solutions, such as: installing portable air cleaners, using ultraviolet lights in the upper room and in air ducts, better filtration systems, enhancing ventilation rates, and using a staggered schedule. This could allow facility managers optimize resources while promoting improved safety. Further, studies like these could also be performed in a variety of congregate settings, such as schools or office buildings, in order to understand the areas of highest risk and similarly inform the effective use of mitigation strategies.

## Supporting information

**S1 Table. Summary of results from the comprehensive air sampling at Site A.** No SARS-CoV-2 was detected in any of the locations sampled. SFTPC was detected in 14 samples distributed across most of the areas. ML = men's locker room, WL = women's locker room and C = cafeteria.
(PDF)

**S2 Table. Summary of results from the comprehensive air sampling at Site B.** No SARS-CoV-2 was detected in any of the locations sampled. SFTPC was detected in 6 samples most of which were in the common areas and cafeterias. ML = men's locker room, WL = women's locker room and C = cafeteria.
(PDF)

**S3 Table. Summary of results from the comprehensive air sampling at Site C.** SARS-CoV-2 was detected in both long-term (1) and discrete term air samples (1). SFTPC was detected in 18 samples distributed across most of the areas. ME = Main Entry, MH = Main Hallway,

ML = men's locker room, WL = women's locker room and C = cafeteria.
(PDF)

## Acknowledgments

The authors would like to thank additional members of the research team who assisted with onsite data collection activities, including Gabriel Lucero, Dr. Ryan Klataske, and Karen Schmeits. The team would also like to thank Dr. Shelly Miller for her guidance on the HVAC assessments.

## Author Contributions

**Conceptualization:** Joshua L. Santarpia, Josephine Lau, Shanna A. Ratnesar-Shumate, John J. Lowe, Athena K. Ramos.

**Formal analysis:** Joshua L. Santarpia, Josephine Lau, Debayan Shom, Shanna A. Ratnesar-Shumate, Vicki L. Herrera, Danielle N. Rivera, Athena K. Ramos.

**Funding acquisition:** Joshua L. Santarpia, John J. Lowe, Athena K. Ramos.

**Investigation:** Joshua L. Santarpia, Josephine Lau, Debayan Shom, Shanna A. Ratnesar-Shumate, Vicki L. Herrera, Danielle N. Rivera, Daniel N. Ackerman, Ashley R. Ravnholdt, John J. Lowe, Athena K. Ramos.

**Methodology:** Joshua L. Santarpia, Josephine Lau, Shanna A. Ratnesar-Shumate, George W. Santarpia, Vicki L. Herrera.

**Supervision:** Joshua L. Santarpia, Josephine Lau, Eric C. Carnes, John J. Lowe.

**Visualization:** Joshua L. Santarpia, Josephine Lau.

**Writing – original draft:** Joshua L. Santarpia, Josephine Lau, Athena K. Ramos.

**Writing – review & editing:** Joshua L. Santarpia, Josephine Lau, Shanna A. Ratnesar-Shumate, Eric C. Carnes, George W. Santarpia, Vicki L. Herrera, Danielle N. Rivera, Daniel N. Ackerman, Ashley R. Ravnholdt, John J. Lowe, Athena K. Ramos.

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
