## [Decision Letter · Decision Letter 0]

10 Sep 2024

PONE-D-24-31372Association between in situ ventilation and human-generated aerosol exposure in meatpacking plants during the COVID-19 pandemicPLOS ONE

Dear Dr. Santarpia,

Thank you for submitting your manuscript to PLOS ONE. After careful consideration, we feel that it has merit but does not fully meet PLOS ONE’s publication criteria as it currently stands. Therefore, we invite you to submit a revised version of the manuscript that addresses the points raised during the review process.

We look forward to receiving your revised manuscript.

Kind regards,

Rajeev Singh

Academic Editor

PLOS ONE

Journal Requirements:

2. Thank you for stating the following financial disclosure: Funding for this study was provided through private donations to the University of Nebraska Foundation.

3. Thank you for stating the following in the Acknowledgments Section of your manuscript: Funding for this study was provided through private donations to the University of Nebraska Foundation. The authors would like to thank additional members of the research team who assisted with onsite data collection activities, including Gabriel Lucero, Dr. Ryan Klataske, and Karen Schmeits. The

team would also like to thank Dr. Shelly Miller for her guidance on the HVAC assessments. Many of the AirAnswers samplers used in this study were a gift from Inspirotec, LLC. 

Please remove any funding-related text from the manuscript and let us know how you would like to update your Funding Statement. Currently, your Funding Statement reads as follows: Funding for this study was provided through private donations to the University of Nebraska Foundation.

4. Thank you for stating the following in the Competing Interests section: Santarpia has been a paid consultant for both Inspirotec and Poppy Health, both of whom develop and provide indoor infectious disease and allergen air monitoring services and devices. All other authors report no conflict. The funders had no role in the design of the study; in the collection, analyses, or interpretation of data; in the writing of the manuscript; or in the decision to publish the results. Many of the AirAnswers samplers used in this study were a gift from Inspirotec, LLC. 

5. We notice that your supplementary tables are included in the manuscript file. Please remove them and upload them with the file type 'Supporting Information'. Please ensure that each Supporting Information file has a legend listed in the manuscript after the references list.

Reviewers' comments:

Reviewer's Responses to Questions

**Comments to the Author**

1. Is the manuscript technically sound, and do the data support the conclusions?

Reviewer #1: Yes

2. Has the statistical analysis been performed appropriately and rigorously? 

Reviewer #1: I Don't Know

3. Have the authors made all data underlying the findings in their manuscript fully available?

Reviewer #1: No

4. Is the manuscript presented in an intelligible fashion and written in standard English?

Reviewer #1: Yes

5. Review Comments to the Author

Reviewer #1: Thank you for this overall nicely written and relevant manuscript. I think it is an approach to discuss which working areas are most exposed to SARS-CoV-2. Find attached some general and specific comments:

1) As someone who has closely followed the outbreak events in meat plants, I realized that in my country meat factories are very heterogeneous depending on the focus. Your sample consists of three facilities from America and I also had the feeling that the plants there are very heterogeneous. Is it possible to generalize the results in this way?

2) Data availability & Methods & Statistical part - you are reporting a lot of parameter like mean, standard deviation, correlation (,etc.?) and for me it was unclear how you calculated these estimates. Could you explain this in more detail in the methods section? Otherwise I can't judge whether you have really disclosed all the data, especially for the mean values.

3) Could you provide more information about the measurements of each of the tests in the methods section? How and when were the measurements carried out? Were all measurements long-term measurements or were individual measurements only carried out once. This would have to be specified here. Were all measurements in real life situations or were some measurements in empty rooms?

4) Something went wrong with the tables and there are two different tables 1. Formatting is also different here. This made it really difficult to follow the text. Table numbering (from Table 4 onwards) is therefore incorrect.

5) In the part "Assay of Samples" I have the impression that methods are here in the result section. Unfortunately, I cannot really assess the results here and if results are presented in this way, the comment can be ignored.

6) I would structure the discussion differently. First summarize the most important results and then in the strengths section you can go into why the study is special. You can also take a look at the recommedations in the strobe statement (https://www.strobe-statement.org/).

7) Source 19 looks a bit strange (Herstein J, SM, LA, DE, RA, GB, B-MD, KC, LJ, & LJ.). Overall, the bibliography should be standardized again.

6. PLOS authors have the option to publish the peer review history of their article (what does this mean?). If published, this will include your full peer review and any attached files.

Reviewer #1: No

---

## [Author Response · Author response to Decision Letter 0]

17 Oct 2024

Response to Reviewers

We appreciate the time and effort by the reviewer to provide comments on our manuscript. We have carefully considered each comment and have provided a point-by-point response (in red) to each comment. We have also included a track changes version of manuscript, along with the clean version. 

Reviewer #1: Thank you for this overall nicely written and relevant manuscript. I think it is an approach to discuss which working areas are most exposed to SARS-CoV-2. Find attached some general and specific comments:

1) As someone who has closely followed the outbreak events in meat plants, I realized that in my country meat factories are very heterogeneous depending on the focus. Your sample consists of three facilities from America and I also had the feeling that the plants there are very heterogeneous. Is it possible to generalize the results in this way?

The reviewer raises an important point regarding the heterogeneity of the meatpacking facilities worldwide. As described in the manuscript, Sites A and C were similar in size, scale and ventilation, while Site B was significantly newer, processed a different protein (poultry, rather than beef) and had much different work practices. The conclusions regarding increased airflow minimizing exposure to human-generated aerosol were generated by synthesizing the data from all 3 facilities. The conclusion that under ventilated spaces represent a greater risk of exposure is not unique to this study, but the data gathered here strongly support that increasing ventilation reduces the exposure to human-generated aerosols that may transmit disease. 

2) Data availability & Methods & Statistical part - you are reporting a lot of parameter like mean, standard deviation, correlation (,etc.?) and for me it was unclear how you calculated these estimates. Could you explain this in more detail in the methods section? Otherwise I can't judge whether you have really disclosed all the data, especially for the mean values.

We originally assumed that the calculated means and standard deviations were sufficient in our data disclosure. Based on the reviewers comments, we are now posting all raw data to the following data repository: Santarpia, Joshua, 2024, "Replication Data for: Association between in situ ventilation and human-generated aerosol exposure in meatpacking plants during the COVID-19 pandemic", https://doi.org/10.7910/DVN/J6BQES, Harvard Dataverse

3) Could you provide more information about the measurements of each of the tests in the methods section? How and when were the measurements carried out? Were all measurements long-term measurements or were individual measurements only carried out once. This would have to be specified here. Were all measurements in real life situations or were some measurements in empty rooms?

Due to the release of higher concentrations of CO2 in Test 1, the test was carried out during no occupancy. Tests 2, 3, 4 and 5 were carried out during normal work, when the spaces were occupied. Test 5, in particular, was carried out during periods of maximum occupancy (page 10 in the test 5 description). Test 1 was performed according to a standard method (ASTM D6245), Test 2 was calculated based on 5 successive measurements (as described in the text, Page 8). As described in the text, the ventilation for Test 3 was calculated based on measurements over 5 consecutive days. The number of individual samples for Tests 4 and 5 are described in Table 2, as well as in Table 5 and all supplemental tables. Test 4 was the only multi-day measurement, where single samples were collected over periods of several days (as described in the text). For tests 3 and 5, short duration measurements were made repeatedly over several days. The descriptions of the tests have been updated (where not explicitly stated) to reflect the above description.

4) Something went wrong with the tables and there are two different tables 1. Formatting is also different here. This made it really difficult to follow the text. Table numbering (from Table 4 onwards) is therefore incorrect.

We appreciate the reviewer identifying this issue. The original manuscript version had many tables in the text that were moved to the supplement during editing. There were some residual incorrect figure references, as well as misnumbered tables that we did not catch prior to submission. The tables have now been correctly numbered and referenced in the text.

5) In the part "Assay of Samples" I have the impression that methods are here in the result section. Unfortunately, I cannot really assess the results here and if results are presented in this way, the comment can be ignored.

The “Assay of Samples” is a subsection of the Methods. It describes, in detail, how each of the samples collected in the preliminary study of Site A and Tests 4 and 5 were assayed for the presence of SARS-CoV-2 and SFTPC, how the copy number is calculated from the raw Ct value, and how the copy number is converted to relevant environmental concentration units, based on how each sample was collected. The resulting calculated data is presented in the Tables and Figures in the text and supplement. 

6) I would structure the discussion differently. First summarize the most important results and then in the strengths section you can go into why the study is special. You can also take a look at the recommendations in the strobe statement (https://www.strobe-statement.org/).

We have worked to restructure the discussion section according to the reviewer’s recommendations. 

7) Source 19 looks a bit strange (Herstein J, SM, LA, DE, RA, GB, B-MD, KC, LJ, & LJ.). Overall, the bibliography should be standardized again.

We thank the reviewer for catching this error. There were some import errors in the reference manager software, which we have now corrected.

---

## [Decision Letter · Decision Letter 1]

15 Nov 2024

PONE-D-24-31372R1Association between in situ ventilation and human-generated aerosol exposure in meatpacking plants during the COVID-19 pandemicPLOS ONE

Dear Dr. Santarpia,

Thank you for submitting your manuscript to PLOS ONE. After careful consideration, we feel that it has merit but does not fully meet PLOS ONE’s publication criteria as it currently stands. Therefore, we invite you to submit a revised version of the manuscript that addresses the points raised during the review process.

We look forward to receiving your revised manuscript.

Kind regards,

Rajeev Singh

Academic Editor

PLOS ONE

Journal Requirements:

Reviewers' comments:

Reviewer's Responses to Questions

**Comments to the Author**

1. If the authors have adequately addressed your comments raised in a previous round of review and you feel that this manuscript is now acceptable for publication, you may indicate that here to bypass the “Comments to the Author” section, enter your conflict of interest statement in the “Confidential to Editor” section, and submit your "Accept" recommendation.

Reviewer #1: All comments have been addressed

2. Is the manuscript technically sound, and do the data support the conclusions?

Reviewer #1: Yes

3. Has the statistical analysis been performed appropriately and rigorously? 

Reviewer #1: Yes

4. Have the authors made all data underlying the findings in their manuscript fully available?

Reviewer #1: Yes

5. Is the manuscript presented in an intelligible fashion and written in standard English?

Reviewer #1: Yes

6. Review Comments to the Author

Reviewer #1: Thank you for incorporating my comments. I have to ask again about point two. Could you include which software was used to perform the calculations? If all the calculations have already been done by the measuring devices, the description of how the tables and figures were created is also sufficient. Finally, I can only find one error in Table 6, where the table labeling is strange.

7. PLOS authors have the option to publish the peer review history of their article (what does this mean?). If published, this will include your full peer review and any attached files.

Reviewer #1: No

---

## [Author Response · Author response to Decision Letter 1]

17 Nov 2024

Response to Reviewers

We appreciate the time and effort by the reviewer to provide comments on our manuscript. We have carefully considered each comment and have provided a point-by-point response (in red) to each comment. We have also included a track changes version of manuscript, along with the clean version. 

Reviewer #1: Thank you for incorporating my comments. I have to ask again about point two. Could you include which software was used to perform the calculations? If all the calculations have already been done by the measuring devices, the description of how the tables and figures were created is also sufficient. Finally, I can only find one error in Table 6, where the table labeling is strange.

We appreciate the reviewers clarification of their previous comment. In response, we have added a short section at the end of the methods section entitled “Data tabulation and Analysis”, which addresses which software packages were utilized, and in what way for the analysis of the data.

We have attempted to clarify the titles in Table 6 to ensure that they are clear.

---

## [Editor Report · Decision Letter 2]

19 Nov 2024

Association between in situ ventilation and human-generated aerosol exposure in meatpacking plants during the COVID-19 pandemic

PONE-D-24-31372R2

Dear Dr. Santarpia,

We’re pleased to inform you that your manuscript has been judged scientifically suitable for publication and will be formally accepted for publication once it meets all outstanding technical requirements.

Kind regards,

Rajeev Singh

Academic Editor

PLOS ONE
---

## [Editor Report · Acceptance letter]

22 Nov 2024

PONE-D-24-31372R2 

PLOS ONE

Dear Dr. Santarpia, 

I'm pleased to inform you that your manuscript has been deemed suitable for publication in PLOS ONE. Congratulations! Your manuscript is now being handed over to our production team.

Kind regards, 

on behalf of

Dr. Rajeev Singh 

Academic Editor

PLOS ONE